

# Coastal Change Patterns from Time Series Clustering of Permanent Laser Scan Data

Mieke Kuschnerus[1], Roderik Lindenbergh[1], and Sander Vos[2]

[1]Department of Geoscience and Remote Sensing, Delft University of Technology
[2]Department of Hydraulic Engineering, Delft University of Technology

**Correspondence:** M. Kuschnerus (m.kuschnerus@tudelft.nl)

**Abstract.** Sandy coasts are constantly changing environments governed by complex interacting processes. Permanent laser scanning is a promising technique to monitor such coastal areas and support analysis of geomorphological deformation processes. This novel technique delivers 3D representations of a part of the coast at hourly temporal and centimetre spatial resolution and allows to observe small scale changes in elevation over extended periods of time. These observations have the potential to improve understanding and modelling of coastal deformation processes. However, to be of use to coastal researchers and coastal management, an efficient way to find and extract deformation processes from the large spatio-temporal data set is needed. In order to allow data mining in an automated way, we extract time series in elevation or range and use unsupervised learning algorithms to derive a partitioning of the observed area according to change patterns. We compare three well known clustering algorithms, k-means, agglomerative clustering and DBSCAN, and identify areas that undergo similar evolution during one month. We test if they fulfil our criteria for a suitable clustering algorithm on our exemplary data set. The three clustering methods are applied to time series of 30 epochs (during one month) extracted from a data set of daily scans covering a part of the coast at Kijkduin, the Netherlands. A small section of the beach, where a pile of sand was accumulated by a bulldozer is used to evaluate the performance of the algorithms against a ground truth. The k-means algorithm and agglomerative clustering deliver similar clusters, and both allow to identify a fixed number of dominant deformation processes in sandy coastal areas, such as sand accumulation by a bulldozer or erosion in the intertidal area. The DBSCAN algorithm finds clusters for only about 44% of the area and turns out to be more suitable for the detection of outliers, caused for example by temporary objects on the beach. Our study provides a methodology to efficiently mine a spatio-temporal data set for predominant deformation patterns with the associated regions, where they occur.

## 1 Introduction

Coasts are constantly changing environments that are essential to the protection of the hinterland from the effects of climate change and, at the same time, belong to the areas that are most affected by it. Especially long-term and small scale processes



prove difficult to monitor but can have large impacts. To improve coastal monitoring and knowledge of coastal deformation processes, a new technique called Permanent Laser Scanning (PLS) (also called continuous laser scanning) based on Light detection and ranging (LiDAR) measurements is available. For this purpose, a laser scanner is mounted on a high building close to the coast in a fixed location acquiring a 3D scan every hour during several months up to years.

The resulting spatio-temporal data set consists of a series of point cloud representations of a section of the coast. The high temporal resolution and long duration of data acquisition in combination with high spatial resolution (in the order of centimetres) provides a unique opportunity to capture a near continuous representation of ongoing deformation processes, like for example storm and subsequent recovery, on a section of the coast. As analysed by Lazarus and Goldstein (2019), the natural effects of a storm on a typical urban beach can rarely be analysed separately from anthropogenic activities, since in most cases work with bulldozers starts immediately after or even during severe storms. There is a need for the detection and quantification of change processes that influence the geomorphology of the coast, to allow understanding and modelling them, as the reaction of the coast to extreme weather events gains importance, Masselink and Lazarus (2019). More examples for potential use of such a data set are presented by O'Dea et al. (2019), who use data from a similar set-up in Duck, USA.

The PLS data set is large (in the order of hundreds of gigabytes), and to be relevant, the information on deformation processes has to be extracted concisely and efficiently. Currently there are no automated methods for this purpose and studies focus on one or a few two dimensional cross-sections through the data (for example O'Dea et al. (2019)). The high temporal resolution and long observation period lead to a data set of high dimensional time series (i.e. 30 epochs up to several thousands). Data mining on high dimensional data sets can be challenging as concluded by Verleysen and François (2005). In a first step towards extraction of interesting events and change patterns we build on the method introduced by Lindenbergh et al. (2019). We use clustering algorithms on time series representing the evolution of the data set, to group them according to their change pattern and then identify underlying processes. We use clustering (or unsupervised learning) to avoid having to specify the patterns and processes that we are looking for in advance.

One example of spatio-temporal segmentation on our data set from permanent laser scanning was recently developed by Anders et al. (2020). They detected seed points for deformation in time series from permanent laser scanning, to then grow a region affected by the detected change around the seed points with the use of dynamic time warping distance to spatial neighbours. Dynamic time warping is a distance measure between time series, that accounts for similarity in patterns even though they might be shifted in time (see for example Keogh and Ratanamahatana (2005)). One drawback of this approach is that temporal patterns of interest have to be defined before hand, and therefore only deformation patterns that are expected can be found. Another approach to model spatio-temporal deformations in point clouds from laser scanning, is presented by Harmening and Neuner (2020). Their model assumes that the deformation can be represented by a continuous B-spline surface. This approach could potentially be used to further analyse some of the deformation patterns found in our study but does not allow the exploratory data mining, that we are aiming to accomplish. A more general overview of methods to find spatio-temporal patterns in earth science data was published by Tan et al. (2001) and a continuation of this study was presented by Steinbach et al. (2001). The study of Tan et al. deals with pre-processing of time series of different variables from satellite data including issues with auto-correlation and seasonality. Steinbach et al. successfully apply a novel clustering technique





introduced by Ertöz et al. (2003) to explore spatio-temporal climate data. However, this technique only focuses on contiguous clusters, where all time series are in a close neighbourhood to each other, and does not allow to find general patterns independent of location.

Time series data sets are also used to asses patterns of agricultural land use by Recuero et al. (2019). For this study time series of Normalized Difference Vegetation Index (NDVI) data have been analysed using auto-correlation values and random forest classification. Benchmark data from an alternative source was needed to train the classifier. Such benchmark data is currently not available in our case. A study by Belgiu and Csillik (2018) used time series from Sentinel-2 satellite data for cropland mapping. They made use of dynamic time warping classification and showed that in areas with little available reference data for training a classifier, their approach delivers good results in segmentation based on time series' evolution. Also in this case pre-labelled training data is required. Another approach using expectation-based scan statistics was presented by Neill (2009): To detect spatial patterns in time series from public health data, a statistical method based on expectation values is used. Clusters are formed where the observed values significantly exceed the expectation. The results are promising but depend on the choice of time series analysis method, statistics used and the shape of the search region, which all have to be defined in advance specific to each data set and application. Generally there is a lack of studies on mining spatio-temporal data for deformation patterns, without using training data or predefined change patterns.

The goal of the present study is to evaluate the application of clustering algorithms on a high dimensional spatio-temporal data set without specifying deformation patterns in advance. Our objectives in particular are:

1. To analyse and compare the limits and advantages of three clustering algorithms for separating and identifying change patterns in high dimensional spatio-temporal data.

2. To detect specific deformation on sandy beaches by clustering time series from permanent laser scanning.

We compare the k-means algorithm, agglomerative clustering and the DBSCAN algorithm on a PLS data set of 30 epochs, to investigate the effectiveness of the identification of coastal change patterns. All three algorithms are well established and represent three common but different approaches to data clustering. To determine if an algorithm is suitable, we expect that it fulfils the following criteria:

– A majority of the observation area is separated into distinct regions,

– each cluster shows a change pattern that can be associated with a geomorphic deformation process, and

– time series contained in each cluster roughly follow the mean change pattern.

Additionally, we compare two representations of time series: First we use time series extracted from a grid in Cartesian coordinates as elevation per grid cell and second time series from a grid in spherical coordinates represented by range per grid cell which is a more native way of representing laser scanner data. We use the different clustering approaches on a small area of the beach at the bottom of a footpath, where sand accumulated after a storm, and a bulldozer subsequently cleared the path and formed a pile of sand. We determine the quality of the detection of this process for both algorithms and compare them





Earth **Surface**
**Dynamics**
Discussions

in terms of standard deviation within the clusters and area of the beach covered by the clustering. We compare and evaluate

the resulting clusters using these criteria as a first step towards the development of a method to mine the entire data set from

permanent laser scanning for deformation processes.

## 2    The permanent laser scan data set

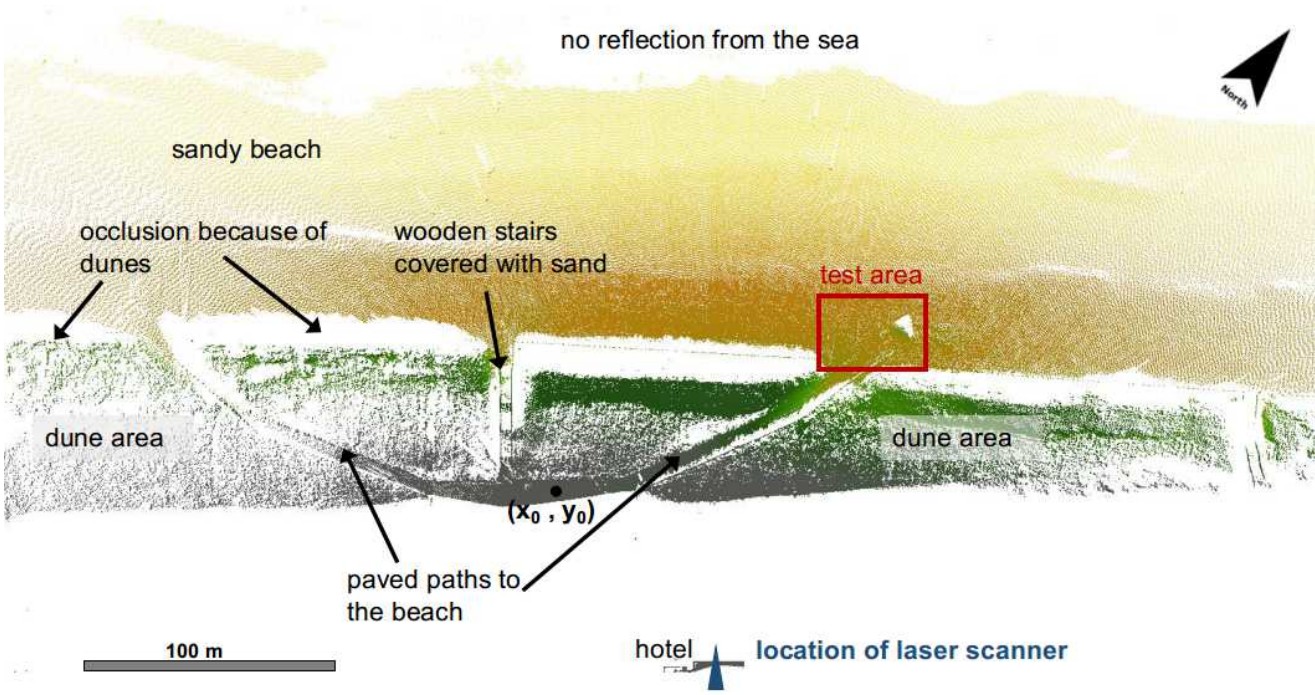

**Figure 1.** Top view of point cloud representing the observation area at low tide on 1st January 2019. Colours represent elevation from low

(yellow, about sea level) to high (grey, about 14 m above sea level and higher). The point $(x_0, y_0)$ indicates the location of the time series

shown as an example in Figure 3. The test area, which is discussed in Section 3.4, is indicated in red at the end of the northern path leading

to the beach.

     The data set from permanent laser scanning is acquired within the CoastScan project at a typical urban beach in Kijkduin,

the Netherlands, Vos et al. (2017). For the acquisition a Riegl VZ-2000 laser scanner was used to scan over a period of six

months from December 2016 to May 2017. The full data set consists of hourly scans of a section of sandy beach and dunes.

     For the present study, a subset of the available data is used to develop the methodology. This subset consists of 30 daily

scans taken at low tide over a period of one month, January 2017. It covers a section of the beach and dunes in Kijkduin and is

displayed in top view in Figure 1. The area contains a path and stairs leading down to the beach, a paved area in front of the

dunes, a fenced in dune area and the sandy beach. It is about 950 m long, 250 m wide and the distance from the scanner to the



Earth **Surface**
Dynamics
Discussions


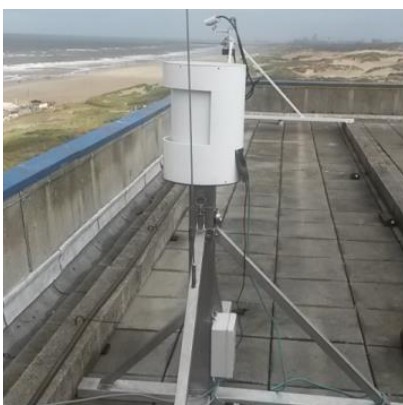

**Figure 2.** Riegl VZ2000 laser scanner mounted on the roof of a hotel facing the coast of Kijkduin, the Netherlands. The scanner is covered with a protective case to shield it from wind and rain.

farthest points on the beach is just below 500 m. For the duration of the experiment the scanner was mounted on the roof of a hotel just behind the dunes at a height of about 37 m above sea level (as shown in Figure 2).

The data is extracted from the laser scanner output format and converted into a file that contains xyz-coordinates and spherical coordinates for each point. The data is mapped into a local coordinate system, where the origin in x- and y-direction is at the location of the scanner and the height (z-coordinate) corresponds to height above sea level. Since we are interested in relative changes between consecutive scans, we do not transform the data into a geo-referenced coordinate system for this analysis.

Each point cloud is chosen to be at the time of lowest tide between 18:00 and 06:00, in order to avoid people and dogs on the beach, with the exception of two days where only very few scans were available due to maintenance activities. The data from 9[th] of January 2017 is entirely removed from the data set, because of poor visibility due to fog. Additionally all points above 14.5 m elevation are removed to filter out points representing the balcony of the hotel and flag posts along the paths that are interfering with the spherical coordinate grid extraction. In this way also a majority of reflections from particles in the air, birds or raindrops are removed. However, some of these particles might still be present at lower heights close to the beach.

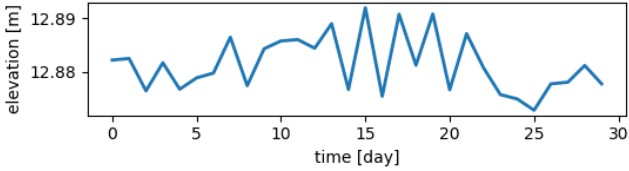

**Figure 3.** Time series in elevation at location $(x_0, y_0)$ (marked in Figure 1) on the path that is assumed to be stable throughout the entire month. Elevation is varying within less than 2 cm.

Since the data is acquired from a fixed and stable position we assume that consecutive scans are aligned. Nevertheless, the orientation of the scanner may change slightly due to strong wind, sudden changes in temperature, or maintenance activities.



The internal inclination sensor of the scanner measures these shifts while it is scanning and we apply a correction for large deviations (more than 0.01 degrees) from the median orientation.

The remaining error in elevation and range is estimated as the average standard deviation of time series in locations that are assumed to be stable during the entire month. We chose stable surfaces that are part of the paved paths on top of the dunes and leading to the beach in northern and southern direction and derived the mean remaining errors shown in Table 1. The elevation

does on average not deviate more then 2 cm from the mean elevation of the respective area and the standard deviation of the range is within 5 cm on the northern path and on top of the dunes, but around 11 cm on the southern path. An example time series from the stable paved area on top of the dunes (at location $(x_0, y_0)$ marked in Figure 1) is shown in Figure 3.

**Table 1.** Average standard deviation of the gridded elevation and range in listed areas, which are each assumed to be stable throughout the observation period of one month.

| mean error | elevation | range |
|---|---|---|
| paved area on top of dunes | 1.47 cm | 4.89 cm |
| path leading to the beach (north) | 1.12 cm | 4.03 cm |
| path leading to the beach (south) | 1.65 cm | 10.9 cm |



## 3   Methods

To derive coastal deformation processes from clusters based on change patterns we follow three steps: Extraction of time series
in two different coordinate systems, clustering of time series with three different algorithms, and derivation of geomorpholog-
ical deformation processes. To cluster time series the definition of a distance between two time series (or the similarity) is not
immediately obvious. We discuss two different options (Euclidean distance and correlation) to define distances between time
series with different effects on the clustering results. The rest of this section is organized as follows: We focus on time series
extraction in subsection 3.1, discuss distance metrics for time series (3.2), introduce three clustering algorithms (3.3) and our
evaluation criteria (3.4). The derivation of deformation processes will be discusses with the results (section 4).

### 3.1   Time Series Extraction

Time series are extracted from the PLS data set in two different ways: First by using a grid in Cartesian xy-coordinates and
extracting time series in elevation and second by using a grid in spherical coordinates and extracting time series in range. For
both methods, we only use grid cells that contain at least one point for each of the scans.

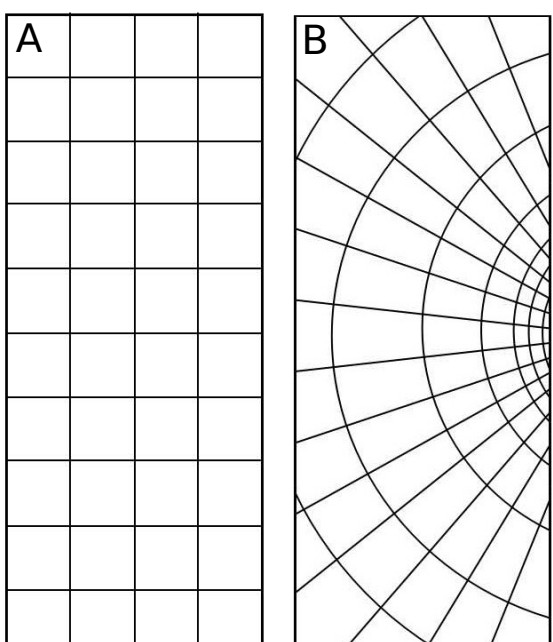

**Figure 4.** Regular grid in Cartesian coordinates (A) and in spherical coordinates (B). In spherical coordinates (B) the size of the grid cells is
larger, the further away they are from the scanner because of the large incidence angle between the line of sight of the scanner and the normal
of the sloping surface of the beach.



Earth **Surface**
**Dynamics**
Discussions



### 3.1.1 Cartesian Coordinates

Before defining a grid in Cartesian coordinates, we rotate the observation area to make sure that the coastline is parallel to the y-axis. This ensures that the grid covers the entire observation area efficiently and leaves as few empty cells as possible. Then we generate a regular grid (as illustrated in Figure 4) with grid cells of 1 m × 1 m. Time series are generated for each grid cell by taking the median elevation $z_i$ for each grid cell and for each time stamp $t_k$. That means, per grid cell with center $(x_i, y_i)$ we have a time series

$$\tilde{Z}_i = (z_i(t_1), \dots z_i(t_T)), \tag{1}$$

with the number of time stamps $T = 30$. To make the time series dependent on change patterns, rather than the absolute elevation values, we remove the mean elevation $\bar{z}_i$ of each time series $\tilde{Z}_i$. This leads to time series

$$Z_i = (\Delta z_i(t_1), \dots, \Delta z_i(t_T), \tag{2}$$

with $\Delta z_i(t_k) := z_i(t_k) - \bar{z}_i$.

This approach results in a collection of time series that represent equally sized grid cells. However, the point density per grid cell varies depending on distance to the laser scanner. For example, a grid cell on the paved path (at about 80 m range) contains about 40 points (i.e. time series at $(x_0, y_0)$ in Figure 1), whereas a grid cell located close to the water line, at about 300 m distance from the scanner, may contain around three values. This implies that the median per grid cell is based on more points the closer a grid cell is to the scanner.

### 3.1.2 Spherical Coordinates

Using spherical coordinates allows to generate a grid with constant angle increment in horizontal and vertical direction with roughly constant point density per grid cell. For each grid cell $j$, we derive a time series $R_j$ consisting of the median range $r_j(t_k)$ per grid cell per time stamp $t_k$:

$$R_j = (r_j(t_1), \dots, r_j(t_T)), \tag{3}$$

where $T = 30$, as above. The range $r_j$ is defined as the line of sight distance from the laser scanner to the respective point. We choose grid cells of $0.1° \times 0.375°$ to ensure that grid cells on the beach (close to the dune foot) cover roughly 1 m², the same as in Cartesian coordinates, in order to make both methods comparable. However, transformed back into Cartesian coordinates, the size of the grid cells (in square-meters) varies with distance from the scanner (see Figure 4).

The point density in a point cloud is generally lower, the farther away a point is from the scanner. This property of our data set is preserved using spherical coordinates and represented in the size of the grid cell, or distance between grid cell centres.

### 3.2 Distance Metrics

We consider two different distance metrics for our analysis.





### 3.2.1 Euclidean Distance

The most common and obvious choice is the Euclidean distance metric defined as:

$$d_E(Z_0, Z_1) = ||Z_0 - Z_1|| = \sqrt{\sum_{i=1}^{n} |Z_{0i} - Z_{1i}|^2}, \tag{4}$$

for two time series $Z_0$ and $Z_1$ of length $n$.

### 3.2.2 Correlation Distance

Another well known distance measure is correlation distance, defined as one minus the Pearson correlation coefficient (see for
example Deza and Deza (2009)). It is a suitable measure of similarity between two time series, when correlation in the data is
expected (see Iglesias and Kastner (2013)). Correlation between two time series $Z_0$ and $Z_1$ is defined as:

$$\text{Cor}(Z_0, Z_1) = 1 - \frac{(Z_0 - \bar{Z}_0) \cdot (Z_1 - \bar{Z}_1)}{||Z_0 - \bar{Z}_0|| \cdot ||Z_1 - \bar{Z}_1||}, \tag{5}$$

with $\bar{Z}$ being the mean value of time series $Z$ and $|| \cdot ||$ the Euclidean 2-norm as in Equation (4). We have to note here, that
correlation cannot compare simple constant time series (leads to division by zeros) and is therefore not a distance metric in the
175 sense of the definition Deza and Deza (2009).

### 3.2.3 Comparison

For a comparison of the two distances for some example time series see Figure 5. The example shows that the distance between
two time series is not intuitively clear. The use of different distance metrics results in different sorting of distances between
the shown pairs of time series. However, when normalizing all time series (subtracting the mean and scaling by the standard
deviation) correlation distance and Euclidean distance are equivalent (as shown for example by Deza and Deza (2009)).

Both Euclidean distance and correlation are not taking into account the order of the values within each time series. For
example, two identical time series that are shifted in time are seen as 'similar' with the correlation distance, but not as similar
with the Euclidean distance and would not be considered as identical by either of them (see Figure 5). Additionally neither of
the two distance metrics can deal with time series of different lengths or containing gaps.




Earth **Surface**
**Dynamics**
Discussions

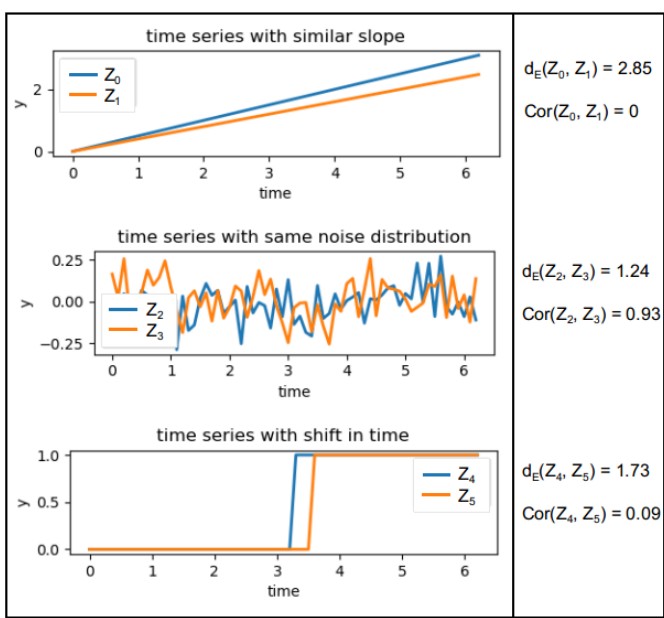

**Figure 5.** Example of three pairs of time series that are 'similar' to each other in different ways. The Euclidean distance would sort the differences as follows $d_E(Z_2, Z_3) < d_E(Z_4, Z_5) < d_E(Z_0, Z_1)$, whereas according to the correlation distance the order would be $\mathrm{Cor}(Z_0, Z_1) < \mathrm{Cor}(Z_2, Z_3) < \mathrm{Cor}(Z_4, Z_5)$.





## 3.3 Clustering Methods

Clustering methods for Time Series can be divided into two categories: feature based and raw data based (see for example Liao (2005)). Feature based methods first extract relevant features to reduce dimensionality (for example using Fourier- or wavelet-transforms) and then form clusters based on these features. They could also be used to deal with gaps in time series. We focus on the raw data based approach to not define features in advance and to make sure that no information within the data set is lost. We use three different methods: k-means clustering, agglomerative clustering and Density-Based Spatial Clustering of Applications with Noise (DBSCAN). In Figure 6 an illustration of a partitioning of a simple 2D data set is shown for each of the three algorithms. The two clusters that can be distinguished in this example have different variances and are grouped differently by each of the algorithms.

For the implementation of all three algorithms, we make use of the Scikit-learn package in Python (see Pedregosa et al. (2011)).

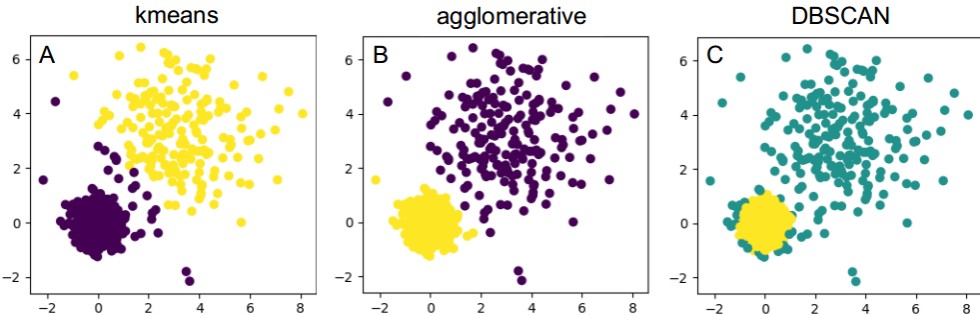

**Figure 6.** Example of clustering of data with two clusters with different variance: The k-means algorithm separates them, but adds a few points in the middle to the purple cluster instead of the yellow one (A). Agglomerative clustering separates both clusters according to their variances (B) and DBSCAN detects the cluster with low variance and high point density (yellow) and discards all other points as outliers (turquoise) (C).





### 3.3.1  k-means Clustering

The k-means algorithm was first introduced in 1955 and is still one of the most widely used clustering methods (Jain (2010)). The algorithm is based on minimizing the sum of all distances between points and centroids over all possible choices of $k$ cluster centroids $V = \{v_1, \ldots, v_k\}$:

$$\text{Min}_V J(V) = \sum_{j=1}^{k} \sum_{x_i \in v_j} ||x_i - v_j||^2, \tag{6}$$

with Euclidean distance metric $|| \cdot ||$. After the initial choice of $k$ centroids among all points the following steps are repeated iteratively, until the above sum does not change significantly:

1. Assign each point to the cluster with closest centroid

2. Move centroid to mean of each cluster

3. Calculate sum of distances over all clusters (Equation (6))

Note that minimizing the squared sum of distances over all clusters, coincides with minimizing the squared sum of all within cluster variances. The convergence to a local minimum can be shown for the use of Euclidean distance (see for example Jain (2010)). The convergence is sped up using so-called k-means++ initialization: After the random selection of the first centroid, all following centroids are chosen based on a probability distribution proportional to their squared distance to the already defined centroids. In this way the initial centroids are spread out throughout the data set and the dependence on the random initialization of the cluster centroids is reduced.

There are variations of k-means using alternative distance metrics such as the $L^1$-norm (k-medoids, Park and Jun (2009)), however the convergence is not always ensured in these cases. Another issue to take into account when considering alternative distance metrics, is the definition of the cluster centroids as mean of time series, which is not automatically defined for any distance metric. For more information on k-means see Jain (2010), Liao (2005) and the documentation of the Scikit-learn package (Pedregosa et al. (2011)).

### 3.3.2  Agglomerative Clustering

Agglomerative clustering is one form of hierarchical clustering: It starts with each point in a separate cluster and iteratively merges clusters together until a certain stopping criterion is met. There are different variations of agglomerative clustering using different input parameter and stopping criteria (see for example Liao (2005) or the documentation of the scikit-learn package (Pedregosa et al. (2011))). We choose the minimization of the sum of the within cluster variances using the Euclidean distance metric (Equation (6), where the centroids $v_j$ are the mean values of the clusters) for a pre-defined number of clusters $k$. The algorithm starts with each point in a separate cluster and iteratively repeats the following steps until $k$ clusters are found:

1. Loop through all combinations of clusters:



– Form new clusters by merging two neighbouring clusters into one

      – Calculate squared sum of distances (Equation (6)) for each combination

   2. Keep clusters with minimal squared sum of distances

In this way we use agglomerative clustering with a similar approach to the k-means algorithm, the same optimization criterion with the same input parameter and Euclidean distance measure. We therefore expect similar results. However, this
agglomerative clustering can easily be adapted to alternative distance measures and could therefore potentially deal with time series of different lengths or containing gaps.

### 3.3.3 DBSCAN Algorithm

Density-Based Spatial Clustering of Applications with Noise, DBSCAN, is a classical example of clustering based on the maximal allowed distance to neighbouring points that automatically derives the numbers of clusters from the data. It was
introduced in 1996 by Ester et al. (1996) and recently revisited by Schubert et al. (2017). The algorithm is based on dividing all points into *core points* or *non-core points* that are close to core points but not themselves surrounded by enough points to be counted as core points. The algorithm needs the maximum allowed distance between points within a cluster ($\varepsilon$) and the minimum number of points per cluster ($N_{min}$) as input parameters. These two parameters define a core point: If a point has a neighbourhood of $N_{min}$ points at $\varepsilon$ distance, it is considered a core point. The algorithm consists of the following steps
(Schubert et al. (2017)):

   1. Determine neighbourhood of each point and identify core points

   2. Form clusters out of all neighbouring core points

   3. Loop through all non-core points and add to cluster of neighbouring core point if within maximal distance, otherwise classify as noise

In this way clusters are formed that truly represent a dense collection of 'similar' points. Since we choose to use correlation as distance metric, each cluster will contain correlated time series in our case. All points that can not be assigned to a close surrounding of a core point, are classified as noise or outliers.

### 3.4 Evaluation Criteria

To determine if an algorithm is suitable, we expect that it fulfils the previously defined criteria:

– A majority of the observation area is separated into distinct regions,

   – each cluster shows a change pattern that can be associated with a geomorphic deformation process, and

   – time series contained in each cluster roughly follow the mean change pattern.





In order to establish these criteria, we compare the three clustering algorithms, as well as the two different ways to derive time series, using the following evaluation methods.

### 3.4.1 Visual Evaluation

The clustered data in Cartesian coordinates are visualized in a top view of the observation area, where each point represents the location of a grid cell and its colour the corresponding cluster, which contains the time series in that location. The centre of each grid cell in spherical coordinates is mapped back to the mean Cartesian xy-coordinates, to visualize the clusters of the range time series in a comparable way. Each cluster is associated with its cluster centroid, the mean time series in elevation of all time series in the respective cluster. In this way, cluster centroids are visualized as elevation time series independent of the coordinate system that was used to generate them. This allows for a direct comparison of the cluster centroids between both time series extraction methods. We subsequently derive change processes visually from the entire clustered area. We establish which kind of deformation patterns can be distinguished and if they match, with what we expect in the respective areas (for example gradual erosion in the intertidal area).

### 3.4.2 Quantitative Evaluation

We use the following criteria to compare the respective clustering and grid generation methods quantitatively:

- percentage of entire area clustered

- minimum and maximum within cluster variation

- percentage of correctly identified change in test area with bulldozer work

The percentage of the area that is clustered differs depending on the algorithm. Especially DBSCAN sorts out points that are too far away (i.e. too dissimilar) from others as noise. This will be measured over the entire observation area. The number of all complete time series counts as 100%.

Each cluster has a mean centroid time series and all other time series deviate from that to a certain degree. We calculate the average standard deviation over the entire month per cluster and report on the minimum and maximum value out of all realized clusters.

### 3.4.3 Test Area

To allow for a comparison of the clusters with a sort of ground truth, we selected a test area at the bottom of the footpath. In this area a pile of sand was accumulated by a bulldozer, after the entrance to the path was covered with lots of sand during the storm, as found by Anders et al. (2019). We chose two time stamps for illustration and show the elevation before the bulldozer activity on 12 January, after the bulldozer activity on 16 January and the difference between the elevations on these two days in Figure 7. The area does not change significantly after this event. Within this test area we classify (manually) each point as 'stable' or 'with significant change' depending on a change in elevation of more than 5 cm (positive or negative). Then we





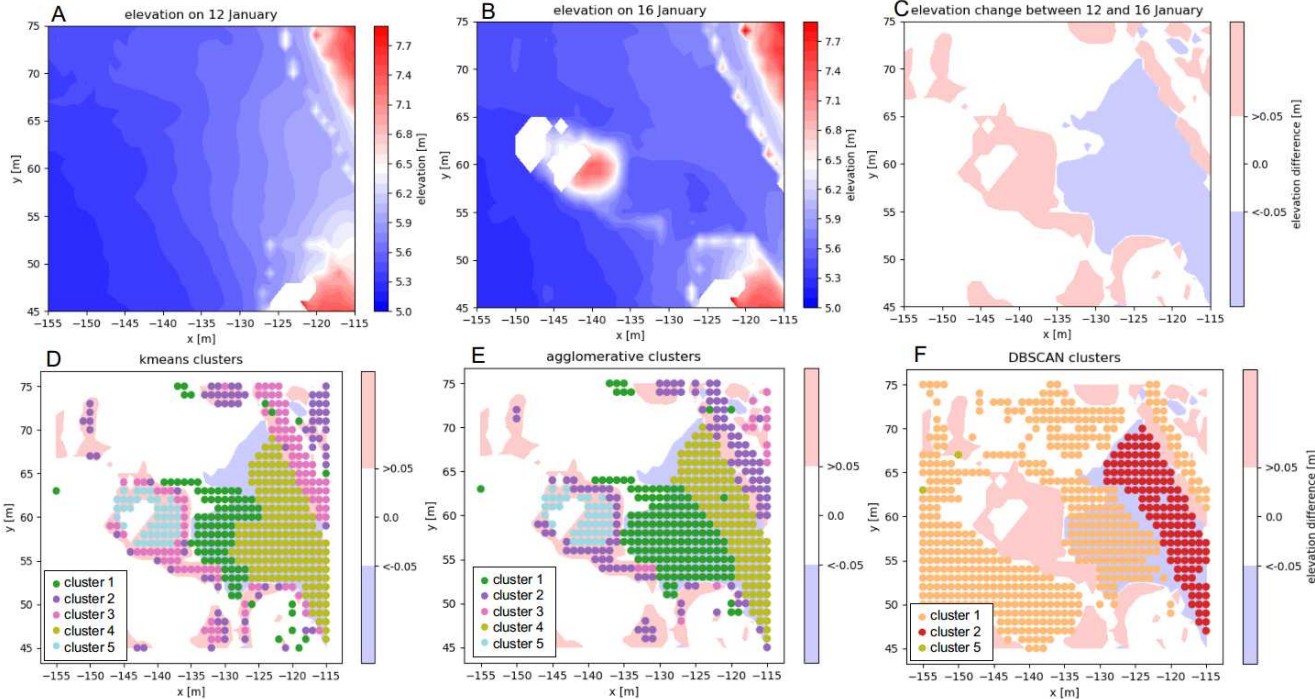

**Figure 7.** Test area for the comparison of clusters generated with different algorithms where the northern access path meets the beach (as shown in Figure 1). 1st row: The elevation in the test area is shown on the day before the bulldozer accumulated a sand pile, when the entrance of the path was covered in sand (A) and after the bulldozer did its job (B). To the right we show the difference in elevation between the two days from a significant level upwards (red) and downwards (blue) (C). 2nd row: Test area with significant changes in elevation and clustered points using the k-means algorithm (D), agglomerative clustering (E) and the DBSCAN algorithm (F). The colours of the clustered dots represent the clusters as shown in Figures 9, 10 and 11, respectively. Points in the 'stable' cluster (cluster 0 for k-means and agglomerative clustering) and outliers (DBSCAN) are not shown.

evaluate for each clustering method if the points that are classified as 'with significant change' are in a separate cluster from the 'stable' points. We do not distinguish if there are different clusters within the category of 'with significant change'. However, in Figure 7, the different clusters can be distinguished by their colours, corresponding to the colours of the clusters shown in subsequent figures (Figures 9, 10 and 11). We then compare the percentage of correctly classified grid points for the test area, for each of the grid generation and clustering methods.





## 4   Results

The results are presented in two parts. First, we compare the different time series extraction methods. Then, we further analyse
the clustering algorithms on the time series in elevation. We compare the results on the test area, where a bulldozer created
a pile of sand (as indicated in Figure 1) and in terms of percentage of data clustered, average standard deviation within each
cluster and physical interpretation of clusters.

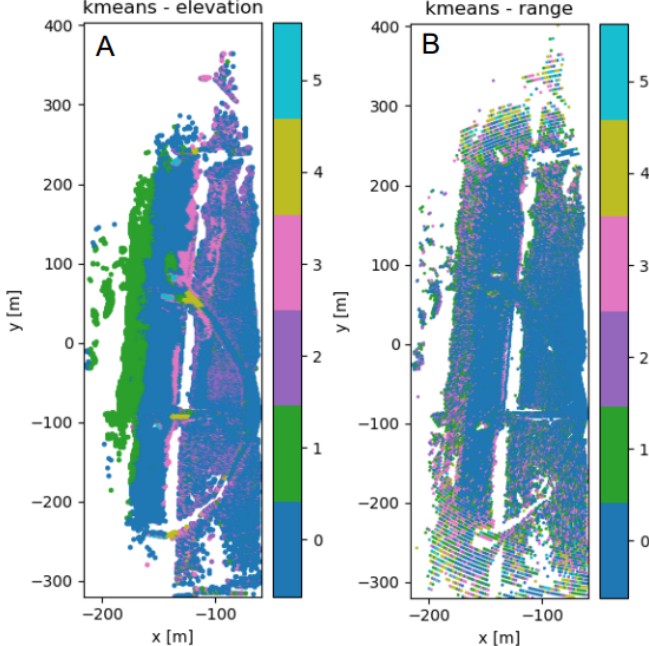

**Figure 8.** Clustered observation area using the k-means algorithm for time series in elevation (A) and time series in range (B). Different
areas are covered and the resulting clusters of range time series appear less suited for identification of change patterns, because the size of
the grid cells is changing depending on the distance to the laser scanner, and deformations in elevation appear less pronounced.

### 4.1   Elevation vs. Range Time Series

With the method to extract time series based on Cartesian coordinates we extract around 40000 grid cells that contain complete
time series in elevation for the entire month. Using the method to extract time series based on spherical coordinates we obtain
around 47000 complete time series in range. In Figure 8 it can clearly be seen that the area covered by both time series
extraction methods is different. The data in spherical coordinates covers a slightly longer part of the beach and more points
at larger distance to the scanner. However, the clusters do not yield a clear partitioning of the beach according to change
patterns. Since the spherical coordinate representation is more native to the scanner and the grid cells generally grow in size





with increasing distance, changes per grid cell over time are less pronounced. Most deformations on the beach are dominated by variations in elevation, which is not captured well in spherical coordinates.

## 4.2   Clustering methods on Elevation Time Series

**Table 2.** Summary of comparison of k-means algorithm, agglomerative clustering (AGG) and DBSCAN algorithm.

|  | k-means | AGG | DBSCAN |
|---|---|---|---|
| entire observation area | | | |
| number of clusters | 6 | 6 | 6 |
| min no. points/cluster | 108 | 108 | 45 |
| area clustered | 100% | 100% | 44% |
| max std within cluster | 3.23 m | 3.19 m | 4.0 m |
| min std within cluster | 0.7 m | 0.72 m | 0.3 m |
| test area: correctly identified | | | |
| stable points | 81% | 86% | 41% |
| positive changes | 97% | 86 % | 1.5% |
| negative changes | 93% | 98 % | 87% |
| total | 85% | 88 % | 45% |

### 4.2.1   K-means

For the k-means algorithm, we choose to use $k = 6$ clusters. From our visual inspection this leads to good, usable results by
partitioning into clusters that are small enough to capture geomorphic changes but not too large, which would make them less informative. With the k-means algorithm, the entire observation area is divided into partitions, which change slightly depending on the random initialization. The standard deviation within each cluster is relatively high, and generally increases with the size of the cluster. Even the cluster with the smallest standard deviation over the entire month, shows a standard deviation of 0.7 m (cluster 5).

On the test area the k-means algorithm correctly classifies about 85% of points into 'stable', 'significant negative change', or 'significant positive change'. However, as can be seen in Figure 7, a part of the points with negative change are not identified. A summary of these results is provided in Table 2.





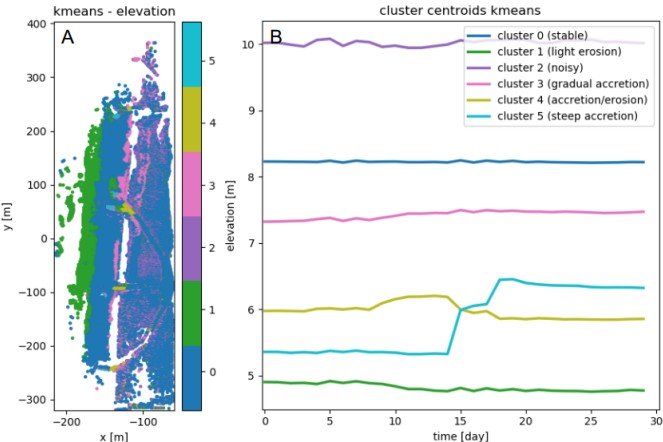

**Figure 9.** A: Overview of the entire observation area divided into six clusters using k-means depending on elevation changes. B: Corresponding cluster centroids for each of the clusters shown in A. The bulldozer activity can be seen between 12 and 16 January in cluster 5.





### 4.2.2 Agglomerative Clustering

The agglomerative clustering algorithm is set up, as the k-means algorithm, to find six clusters. It produces results very similar

to the clusters found with the k-means algorithm, as can be seen comparing Figures 9 and 10 and Figures 7 D and E. Clusters 2 and 3 from agglomerative clustering correspond roughly to the clusters 3 and 2 from k-means clustering. The ordering of clusters is according to size, so more time series are considered 'noisy' according to k-means, whereas agglomerative clustering assigns more of these time series to the gradually accreting cluster. All other clusters appear to be nearly identical and show similar spatial distributions as well as centroid shapes.

On the test area, the detection of negative and positive changes is more balanced and leads to an overall score of 88 % correctly identified points. Agglomerative clustering clearly separates the path that was cleared by the bulldozer and identifies it as eroding.

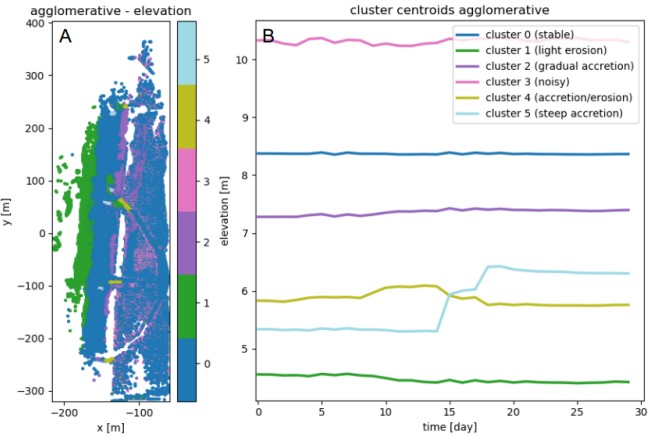

**Figure 10.** A: Overview of the entire observation area divided into six clusters depending on elevation changes using agglomerative clustering. B: Corresponding cluster centroids for each of the clusters shown in A. The clusters are very similar to the ones found with k-means.





### 4.2.3 DBSCAN

When we use the DBSCAN algorithm on the same data set, with minimum number of points $N_{min} = 30$ and maximum distance
$\varepsilon = 0.05$, a large part of the time series (55 %) is classified as noise, meaning that they are not very similar (i.e. not correlated,
since we use correlation as distance measure) to any of the other time series. However they roughly match the combined areas
that are identified as stable and noisy by k-means (clusters 0 and 2). The remaining time series are clustered into six clusters.
The standard deviation within each cluster is generally lower than in the clusters generated with k-means (minimum standard
deviations is 0.4 m) without considering the time series that are classified as noise.
The intertidal zone cannot be separated clearly from the 'noise' part of the observation area, nor can we distinguish the stable
path area or the upper part of the beach. But, two clusters represent areas, which are relatively stable throughout the month,
except for a sudden peak in elevation on one day. These peaks are dominated by a van parking on the path on top of the dunes
and people passing by and not caused by actual deformation in the observed area, as shown in Figure 11.

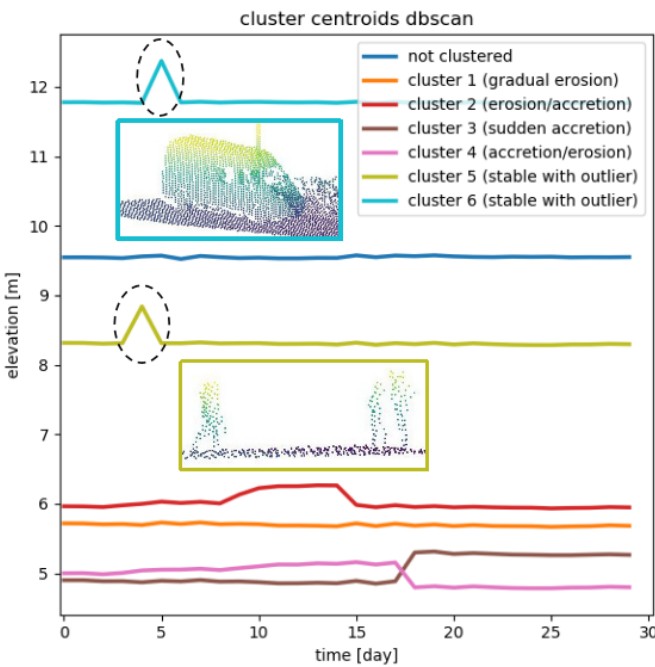

**Figure 11.** Mean time series per cluster found with the DBSCAN algorithm. Outliers or not clustered points are represented by the blue
mean time series. The two most prominent time series (cluster 5 and 6) are located on the path on top of the dunes. The peaks are caused by
a group of people passing by and a van on the 5th and 6th of January respectively, as shown in the point clouds in the middle of the plot.





On the test area the DBSCAN algorithm performs worse than both other algorithms. In total only 45% of points are correctly
classified into 'stable', 'significant negative change', or 'significant positive change'. As stable points we count in this case all
points that are classified as noise, because only time series that show coherent change patterns are clustered by the DBSCAN
algorithm. This matches with about 41% of points classified as stable in the ground truth data. However only 1.5% of the points
with positive significant changes are correctly identified, they are mixed up with a large class of only slightly varying points.
We can see in Figure 7, that a significant part of the stable area is also included in the same cluster.

## 4.3  Identification of Change Processes

From the clustering using range time series, no clear change processes can be distinguished and the beach cannot be partitioned
according to deformation patterns.

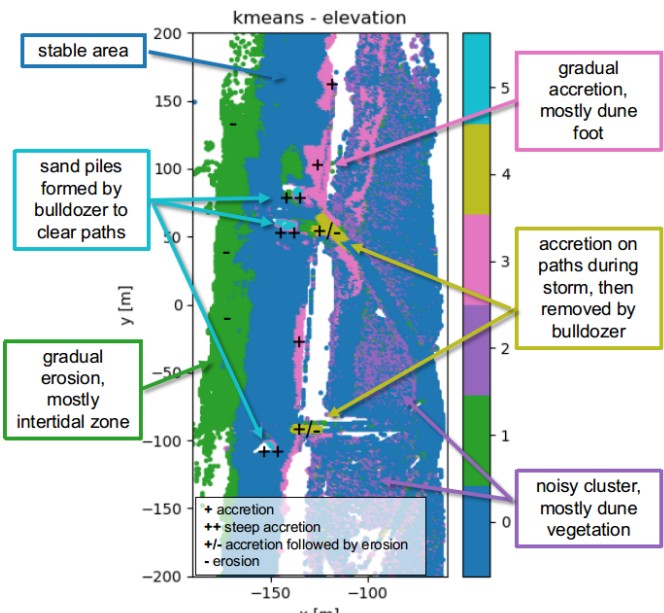

**Figure 12.** Observation area partitioned into clusters with the k-means algorithm used on elevation time series. The blue area represents
the part where time series are mostly stable and show very few change. For the remaining clusters the main cause of the change process is
indicated in the figure. Areas with erosion or accretion are marked with a '-' or '+' symbol. The '++' symbol indicates a steep accretion, and
'+/-' indicates an area where the accretion was followed by erosion (or removal) of sand.

Considering the clusters found by the k-means algorithm and agglomerative clustering on elevation time series, we can
clearly distinguish between time series that represent erosion and gradual accretion, as well as a sudden jump in elevation,
caused by bulldozer work. In Figure 12 we show the clusters and associated main cause for deformations. The cluster dominated
by erosion is close to the water line and roughly represent the inter-tidal zone of the beach and is likely caused by the effects of





tides and waves. The slowly accreting area is mostly at the upper beach, close to the dune foot and likely dominated by aeolian sand transport. The most obvious change process, is the sand removed from the entrances of the paths leading to the beach by bulldozer works (cluster 4) and accumulated in a pile of sand (cluster 5). The noisy cluster is spread out through the dune area
and probably caused by moving vegetation.

## 5 Discussion

We successfully applied the presented methods on a data set from permanent laser scanning and demonstrated the identification of deformation processes from the resulting clusters. Here we discuss our results on time series extraction, distance measures, clustering methods and the choice of their respective input parameters and derivation of change processes.

### 5.1 Time Series Extraction

We compared two different methods to extract time series from the PLS data set either in elevation or in range. The time series extraction in range, which is a more native way of using the data, is very sensitive to vertical structures in the data set, or points in the air in-between the observed surface and the scanner. After removing those points and using the median range per grid cell in spherical coordinates, the time series appear to be dominated by noisy fluctuations, which do not vary a lot depending
on location. Clear change patterns can therefore not be distinguished with any of our algorithms and the distinction of areas that follow a certain change pattern is not possible. A likely cause of this issue, is that most changes are observed in elevation (z-direction) and not in the direction of the range, which makes them less pronounced in spherical coordinates. An alternative approach could be the use of spherical coordinates for the generation of the grid cell, but extraction of time series in elevation instead of range.

In contrast, the time series extraction in Cartesian coordinates provides promising results. Some data is lost, due to low point density in grid cells that are at large distances of the laser scanner. Besides that, the resulting clusters clearly follow recognizable deformation patterns and the clustering allows to separate regions according to these patterns.

### 5.2 Distance Measures

Possible distance measures for the use in time series clustering are analysed among others by Iglesias and Kastner (2013)
and Liao (2005). We use Euclidean distance in combination with the k-means algorithm and agglomerative clustering for our analysis. It has been shown by Keogh and Kasetty (2003) that especially for time series with high dimensions, alternative distance measures rarely outperform Euclidean distance. However, we have to note here, that Euclidean distance is affected by the so called 'curse of dimensionality', which causes a space of time series with many epochs (dimensions) to be difficult to cluster. For more details on this issue see Assent (2012) and Verleysen and François (2005). This could possibly lead to
difficulties, when extending these methods to the use of longer time series, but does not appear to degrade results on our current data set.





We chose for the use of correlation distance with the DBSCAN algorithm, because correlation in principle represents a more intuitive way of comparing time series (see Figure 5). DBSCAN is based on identifying the clusters of high density, which in our case works better using correlation distance instead of Euclidean distance.

380 Neither of the the two distance measures analysed here can deal with gaps in the time series. They also do not allow to identify identical elevation patterns that are shifted in time as similar. An alternative distance measure suitable to deal with these issues would be Dynamic Time Warping (DTW), which accounts for similarity in patterns even though they might be shifted in time (Keogh and Ratanamahatana (2005)). An interpolation method to fill gaps in elevation over short time spans based on surrounding data or a feature based clustering method could be other alternatives.

385 ## 5.3 Clustering Methods

The use of k-means clustering on elevation time series from the same data set was demonstrated by Lindenbergh et al. (2019) and has shown promising first result. We follow the same approach and, as a comparison, use agglomerative clustering, with the same optimization criterion, distance metric and input parameter. As expected the results are similar, although agglomerative clustering does not depend on random initialization. It therefore delivers the same result for every run, which is an advantage.
390 Considering our previously defined criteria:

- a majority of the observation area is separated into distinct regions,

- each cluster shows a change pattern that can be associated with a geomorphic deformation process, and

- time series contained in each cluster roughly follow the mean change pattern,

both algorithms are suitable and the differences in the resulting clusters are negligible for our specific data set.

395 However, the computational effort needed to loop through all possible combinations of merging clusters for agglomerative clustering is considerably higher. Of the three algorithms that were used in this study, agglomerative clustering is the only one that regularly ran into memory errors. This is a disadvantage considering the possible extension of our method to a data set with longer time series.

One of the disadvantages of the k-means algorithm and our configuration of agglomerative clustering, is that the number of
400 clusters has to be defined in advance. Our choice of $k = 6$ clusters yields promising results, but remains somewhat arbitrary, especially without prior knowledge of the data set. We have considered two different methods to determine a suitable value for $k$: analysis of the overall sum of variances for different values of $k$ and so-called *cluster balance* following the approach of Jung et al. (2003). Neither of them resolved the problem satisfactorily and we cannot make a generalized recommendation for the choice of $k$ at this point.

405 To avoid this issue we also compare both approaches with the use of the DBSCAN algorithm. It is especially suitable to distinguish anomalies and unexpected patterns in data as demonstrated by Çelik et al. (2011) using temperature time series. To decide, which values are most suitable for the two input parameters of the DBSCAN algorithms we plot the percentage of clustered points and the number of clusters depending on both parameters (see Figure 13). However, this did not lead to a





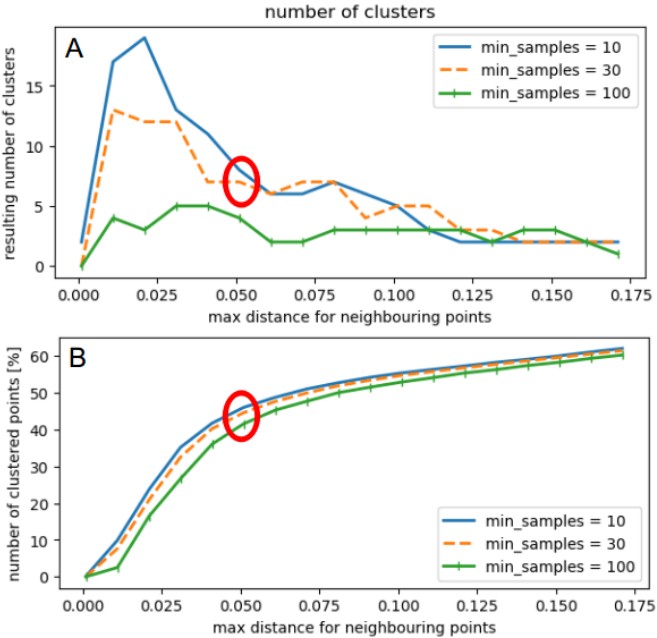

**Figure 13.** DBSCAN selection of input parameters: Number of clusters versus input parameter maximum distance within clusters and minimum number of points and percentage of total points in clusters (not classified as noise/outliers). The choice of an 'optimal' set of parameters is not obvious. We indicate our selection with a red circle in both plots.

clear indication of an 'optimal' set of parameters. After the trade-off analysis between the number of points in clusters and

the number of clusters (not too high, so that the clusters become very small and not too low so that we generate only one big cluster) we chose $\varepsilon = 0.05$ and $N_{min} = 30$ by visually inspecting the resulting clusters.

An alternative clustering approach for time series based on fuzzy C-means is proposed by Coppi et al. (2010). They develop a method to balance the clustering based on the pattern of time series while keeping an approximate spatial homogeneity of the clusters. This approach was successfully applied to time series from socio-economic indicators and could be adapted for

our purpose. It could potentially improve detection of features like sand bars, or bulldozer work, but not distinguish moving vegetation in the dunes as our current approach does.

A similar approach would be to use our clustering results and identified change patterns as input to the region-growing approach of Anders et al. (2020). In this way we could combine advantages of both approaches by making the identification of the corresponding regions for each distinct deformation pattern more exact, without having to define possible deformation

patterns in advance.





## 5.4 Derivation of Change Processes

As shown in Figure 12, we identified change processes from the clusters generated by k-means and agglomerative clustering. Each centroid representing the mean time series of its cluster shows a distinct change pattern (see Figures 9 and 10), which allows to conclude on a predominant deformation. By associating the centroids with the location and spatial spread of the clusters, we can derive the main cause for this deformation. In some cases extra information, or an external source of validation data would be useful to verify the origin of the process. This will be taken into account for future studies. However, the steep rise of the centroid from cluster 5 allows to conclude that the cause of this sudden accretion is not natural. The information found by Anders et al. (2019) for the research on their study, confirms the coinciding bulldozer works.

The DBSCAN algorithm successfully identifies parts of the beach that are dominated by a prominent peak in the time series (caused by a van and a small group of people). Out of the three algorithms that we compare, it is most sensitive to these outliers in the form of people or temporary objects in the data. It was not our goal for this study, to detect people or objects on the beach, but this ability could be a useful application of the DBSCAN algorithm to filter the data for outliers in a pre-processing step.



# 6 Conclusions

We compared three different clustering algorithms (k-means, agglomerative clustering and DBSCAN) on a subset of a large time series data set from permanent laser scanning on a sandy urban beach. We successfully separated the observed beach and dune area according to their deformation patterns. Each cluster, described by the mean time series, is associated with a specific process (such as bulldozer work, tidal erosion) or surface property (for example moving vegetation cover).

The most promising results are found using k-means and agglomerative clustering, which both correctly classify between
440 85 and 88 % of time series in our test area. However, they both need the input of the number of clusters we are looking for and agglomerative clustering is computationally expensive. DBSCAN turned out to be more suitable for the identification of outliers or unnatural occurring changes in elevation due to temporary objects or people in the observed area.

Our key findings are summarized as follows:

1. Both k-means and agglomerative clustering fulfil our criteria for a suitable method to cluster time series from permanent
laser scanning.

2. Predominant deformation patterns of sandy beaches are detected automatically and without prior knowledge using these methods.

3. Change processes on sandy beaches, which are associated with a specific region and time span, are detected in a spatio-temporal data set from permanent laser scanning with the presented methods.

Our results demonstrate a successful method to mine a spatio-temporal data set from permanent laser scanning for predominant change patterns. The method is suitable for the application in an automated processing chain to derive deformation patterns and regions of interest from a large spatio-temporal data set. It allows such a data set to be partitioned in space and time according to specific research questions into phenomena, such as for example the interaction of human activities and natural sand transport during storms, recovery periods after a storm event or the formation of sand banks. The presented methods en-
able the use of an extensive time series data set from permanent laser scanning to support the research on long-term and small scale processes on sandy beaches and improve analysis and modelling of these processes. In this way we expect to contribute to an improved understanding and managing of these vulnerable coastal areas.



*Data availability.* The data set used for this study is available via 4TU Centre for Research Data: https://doi.org/10.4121/uuid:409d3634-0f52-49ea-8047-aeb0fefe78af (Vos et al. (2020)).

*Author contributions.* M. Kuschnerus has carried out the investigation, developed the methodology and software and has realized all visualizations and written the original draft. R.C. Lindenbergh supervised the work and contributed to the conceptualization and to the writing by reviewing and editing. S. Vos developed the instrumental set-up of the laser scanner and collected the raw data set that was used for this research.

*Competing interests.* The authors declare that they have no conflict of interest.

*Acknowledgements.* This work is part of the Open Technology Programme with project number 16352, which is (partly) funded by the Dutch Research Council (NWO). The authors would also like to thank M.C. Mulder for the work on his Bachelor thesis with the title 'Identifying Deformation Regimes at Kijkduin Beach Using DBSCAN'.



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
