# Peer review of "Coastal Change Patterns from Time Series Clustering of Permanent Laser Scan Data"

_Earth Surface Dynamics, 2020_

## Referee Comment (RC1) · Anonymous Referee #1 · 8 Jul 2020

The paper presents a clustering analysis of time series of terrestrial LiDAR in a coastal environment. The paper is well-written and presented. I was able to follow the methodology used by the authors and to understand their conclusions. Since I'm not an expert in time series I recommend minor revisions, mainly about the graphical presentation of the results, but I must also warn to the Editor that the other(s) referree(s) might have a different opinion.

Figure 1 could use a pair of axis to help the reader locate him/herself when referring to figures 8,9,10,12. Also, in the text you mention that the origin of the local coordinate system is at the laser scanner location, but in the figure there is an (xo,yo) just next to the wooden stairs. Which is the correct origin. Also note that in the figure caption you mention 2019 as the scan date, but in the text (line 98) it is 2017.

[Figure]

Figure 2 should be plotted with the same aspect as the test area shown in fig.1 (abaout 40x55m?)

Figures 8,9,10,12 - I found a bit hard at first to relate this with figure 1 (due the lack of axes), so in fig 8 maybe include also the view of fig.1? Perhaps indicate the same features of fig.1 (stairs, road,etc)? Also, did you checked if the colors are safe for colorblind people? If not I suggest a small application called ColorOracle (free, multiplatform) that allows you to simulate the three main colorblindness

In fig.9, try to move the legend so it won't cover any lines, and consider using no only colors, but also different line widths and symbols (dash, dot-dash, etc) so it will be easier to differentiate the lines visually

---

## Referee Comment (RC2) · Anonymous Referee #2 · 21 Oct 2020

This paper presents informative results of a study to automatically differentiate geomorphic surface change dynamics on a coastal back beach. The analysis is comprehensive and robust and the findings are well presented. The manuscript requires some minor revisions before it can be accepted, but I would also urge the authors to better highlight (if possible) what exactly the advantages are of applying a clustering technique. I actually found the results in the end rather underwhelming, because if you remove the two clusters associated with the bulldozer works the results really just yield four different trends of geomorphic change: stable surfaces, steadily eroding and steadily accreting surfaces, and fluctuating surfaces (noise). These four trends are hardly surprising and can just as easily be detected from simple erosion/deposition mapping. The limitation of the application here may be related to the fact that the anal-

ysis was directed to 6 clusters. The manuscript does not give objective or quantitative justification for this decision, other than that the results with 6 clusters seemed good or reasonable to the authors (L304), but this then effectively preempts the possibility of finding something new or interesting. Maybe 10 clusters could have revealed some more interesting trends, for example.

Below follows a list of issues and queries that can be resolved in a minor revision.

L39: define what you mean with "epochs" L39: why "high-dimensional"? This is just a 4D dataset. At this point in the paper it is not clear to the reader yet that you are going to define your data in a multi-dimensional space with the dimensions defined by the snapshots (epochs?). L40: citation here is 15 yrs old, can you refer to more recent literature on the challenges of data mining? L85: why the second representation in terms of range data? The fact that it's the native data format doesn't really give us an actual justification. Your second criterion is about geomorphic deformation, which presumably relates specifically to height changes, hence the cartesian grid seems most suitable to that. Your results later on essentially show that the range format is simply not useful, so you could achieve a great simplification and a more focused message here if you strip out all this stuff about the range format and just report the results related to the cartesian data... Fig.1 says data from 2019, but text says data was from Dec-2016 to May-2017? L122 and earlier: not suitable to use $x_0$ and $y_0$ for identifying a test location as it has nothing to do with zero. Suggest subscript 't' or even 'test'. Table 1 and associated text: we really need more info on these test areas: are they single points? areas? If latter, what size. Then, the test statistics is not informative. Stdev is not sufficient, you should be able to calculate the standard error and the associated 95% confidence intervals around the mean height. Also, why the difference in N vs S? This requires discussion. In the manuscript you present these test results here, but in the results and discussion there should be further reflection on the potential impact of the error on the cluster classifications. L145: you remove the mean in the cartesian format, but not in the range format, why do this in the cartesian grid?

The same logic you use there should somehow apply to the range time-series? More crucially however, in the later results it seems as if the mean was in fact NOT removed, for example in figure 9b, the centroids all have distinct absolute elevations, surely this can only be possible if the mean was not removed? Otherwise the centroids should all be fluctuating around zero? Eq 2 and L145: notation is not suitable; delta usually refers to a real discreet difference; suggest using prime ' as the fluctuating component. Section 3.2: from later on in the results I get the impression that euclideanw as used for k-means and aggregation, while correlation distance was only used for DBSCAN, si this correct? If so, this needs to be stated here. L179-180: then why don't you standardize your data? L245: but you also evaluate Euclidean distance in DBSCAN? (or not, see above?) so what are the clusters in that case? Figure 7: something is seriously going wrong at the white polygons left of centre. The height changes don't match at all. In A the original elevation in this area is around 5.6 m, in B those two white polygons appear to be at 6.4, so this should yield a height change of 0.9 (red) in C. Or are these polygons areas with No Data? If so the colour scales need to be completely different so as to avoid white being part of the scale (so that it can then indicate no-data). Fig7c: show contour lines of height changes beyond significant (as basis for additional clusters?) Stable points not shown? But not clear then how much of area has been allocated properly? L304: why six? Not enough justification. What is 'good'? Why not 8 or 10? Isn't there a statistic to tell you when to stop clustering? Figure 9b: needs horizontal gridlines; labels should be added to lines, rather than a legend (because the sequence in the legend doesn't match the sequence from top to bottom). Vertical axis labels don't make sense: I don't understand how you can have real values asl here for the centroids when the original time-series was mean-subtracted? L341: this is the only place where you really say that there are no results like 4.2 for the range data; this needs to be made more explicit in Line ~300

L372-374: please elaborate a bit on this here, please summarize or give us a taste of the cause for this 'curse'. 5.2: it only gets clear to me here that you use Euclidean for k-means and aggregation, and corr for dbscan! Conclusion: not really clear what all

this work benefits; if you remove the clusters associated with the bulldozer work you basically end up with 4 obvious trends: stable, erosion, accretion, noise. This is a bit underwhelming. . .
* * *

---

## Editor Comment (EC1) · Andreas Baas (Editor) · 21 Oct 2020

Dear Authors, First please accept my sincere apologies for the very long delay on securing review reports of your manuscript. Over the summer months it proved exceedingly difficult to find willing referees, even after invitations to 10 experts around the world (and one false start).

However, we now have two reports that both recommend minor revisions and so I can now invite you to respond to the referee comments and submit a suitably revised manuscript together with an author-response document.

With kind regards, Andreas Baas

---

## Author Comment (AC1) · 16 Nov 2020

Dear referees,

thank you very much for you comments and valuable feedback. Please find below some answers to your comments. More details will be provided as a revised version of the manuscript.

Anonymous Referee #1 Figure 1 could use a pair of axis to help the reader locate him/herself when referring to figures 8,9,10,12.

– Figure 1 will be updated to match the orientation and axis of the following figures for easier comparison. (see attachment)

[Figure]

Also, in the text you mention that the origin of the local coordinate system is at the laser scanner location, but in the figure there is an (xo,yo) just next to the wooden stairs. Which is the correct origin.

– The location of the stable time series will be marked with $(x_t, y_t)$ instead of $(x_0, y_0)$. With the added axes it will be easy to understand that the origin of the coordinate system is at the location of the laser scanner.

Also note that in the figure caption you mention 2019 as the scan date, but in the text (line 98) it is 2017.

– This is a typo in the caption and will be fixed.

Figure 2 should be plotted with the same aspect as the test area shown in fig.1 (abaout 40x55m?)

– Does this refer to Figure 7? Figure 2 is a photo of the laser scanner. I assume this is fixed with the update of Figure 1.

Figures 8,9,10,12 - I found a bit hard at first to relate this with figure 1 (due the lack of axes), so in fig 8 maybe include also the view of fig.1? Perhaps indicate the same features of fig.1 (stairs, road,etc)?

– see above.

Also, did you checked if the colors are safe for colorblind people? If not I suggest a small application called ColorOracle (free, multiplatform) that allows you to simulate the three main colorblindness.

– Thank you for the suggestion. We will look into it.

In fig.9, try to move the legend so it won't cover any lines, and consider using no only colors, but also different line widths and symbols (dash, dot-dash, etc) so it will be easier to differentiate the lines visually

**ESurfD**
– This will be fixed by annotating each curve individually instead of showing a legend, as suggested by Referee #2.

Anonymous Referee #2

[. . .] I would also urge the authors to better highlight (if possible) what exactly the advantages are of applying a clustering technique. I actually found the results in the end rather underwhelming, because if you remove the two clusters associated with the bulldozer works the results really just yield four different trends of geomorphic change: stable surfaces, steadily eroding and steadily accreting surfaces, and fluctuating surfaces (noise). These four trends are hardly surprising and can just as easily be detected from simple erosion/deposition mapping.

– The advantages of our method have to be highlighted and explained in more detail. We will adress this in the result section 'Identification of Change Processes'. Shortly summarized we would like to emphasize the following points:

* Erosion/accretion is detected at different rates, without prior specification of any rate of change or threshold/ We will highlight the specific average rates for different clusters.

* The intertidal area can clearly be distinguished, as well as the location of an 'edge' along the beach between the dry part of the sand and the wet intertidal area.

* Identification of 'noisy' areas in the dunes, which are areas dominated by moving vegetation.

* Possibility to detect the time and location of anthropogenic changes like the sand pile in the test area

The limitation of the application here may be related to the fact that the analysis was directed to 6 clusters. The manuscript does not give objective or quantitative justification for this decision, other than that the results with 6 clusters seemed good or reasonable to the authors (L304), but this then effectively preempts the possibility of finding something new or interesting. Maybe 10 clusters could have revealed some more interesting

trends, for example.

– It is hard to give a general rule for the 'best' choice of the number of clusters k. We will address this in more detail and present different options for the choice of k and their advantages and disadvantages. It comes down to a trade-off between very detailed small clusters, which for example allow to derive the different days, on which the sand piles were erected, versus splitting up a generally stable cluster into small clusters, which does not give new insight into any processes.

L39: define what you mean with "epochs" L39: why "high-dimensional"? This is just a 4D dataset. At this point in the paper it is not clear to the reader yet that you are going to define your data in a multi-dimensional space with the dimensions defined by the snapshots (epochs?).

– The number of dimensions and epochs both refer to the number of time steps in each time series of the data set. I will reformulate to make this more clear.

L40: citation here is 15 yrs old, can you refer to more recent literature on the challenges of data mining?

– yes

L85: why the second representation in terms of range data? The fact that it's the native data format doesn't really give us an actual justification. Your second criterion is about geomorphic deformation, which presumably relates specifically to height changes, hence the cartesian grid seems most suitable to that. Your results later on essentially show that the range format is simply not useful, so you could achieve a great simplification and a more focused message here if you strip out all this stuff about the range format and just report the results related to the cartesian data.

– This is a valid point. We will remove the part on spherical representation and clustering of range time series in order to give the paper more focus.

Fig.1 says data from 2019, but text says data was from Dec-2016 to May-2017?

– This is a typo and will be adapted.

L122 and earlier: not suitable to use x_0 and y_0 for identifying a test location as it has nothing to do with zero. Suggest subscript 't' or even 'test'.

– We will adapt this according to your suggestion.

Table 1 and associated text: we really need more info on these test areas: are they single points? areas? If latter, what size. Then, the test statistics is not informative. Stdev is not sufficient, you should be able to calculate the standard error and the associated 95% confidence intervals around the mean height. Also, why the difference in N vs S? This requires discussion. In the manuscript you present these test results here, but in the results and discussion there should be further reflection on the potential impact of the error on the cluster classifications.

– This is a good point. The stable reference surface can be represented by all paved paths combined. We will provide more information on the statistical properties to emphasize the stability and the order of magnitude of errors. The effect of errors in the instrument on the clustering results is assumed to be negligible, but with a more detailed specification of the rate of change in elevation in the eroding and accreting areas, this will be more straight forward to show. We will add this to the discussion section.

L145: you remove the mean in the cartesian format, but not in the range format, why do this in the cartesian grid? The same logic you use there should somehow apply to the range time-series? More crucially however, in the later results it seems as if the mean was in fact NOT removed, for example in figure 9b, the centroids all have distinct absolute elevations, surely this can only be possible if the mean was not removed? Otherwise the centroids should all be fluctuating around zero?

– The first point is obsolete, since we no longer report on the range time series. (Just for information, removing the mean did not change the results significantly in this case.) We did remove the median elevation for all time series to perform the clustering. In

the later figures the time series that represent the cluster centroids are shown with the median added for visualization purposes. This was not very clear and will be explained.

Eq 2 and L145: notation is not suitable; delta usually refers to a real discreet difference; suggest using prime ' as the fluctuating component.

– Ok, will be adapted according to your suggestion.

Section 3.2: from later on in the results I get the impression that euclidean was used for k-means and aggregation, while correlation distance was only used for DBSCAN, si this correct? If so, this needs to be stated here.

– Yes, we will add some explanation.

L179-180: then why don't you standardize your data?

– Standardizing would make the two distances more comparable, but it does not improve our results. Running the k-means algorithm on standardized time series leads to a separation into very similar relatively stable clusters and does not detect any of the processes (sand pile, erosion, inter-tidal zone) that we find without standardization.

L245: but you also evaluate Euclidean distance in DBSCAN? (or not, see above?) so what are the clusters in that case?

– A short explanation of this case (and why the results are not as good) will be added.

Figure 7: something is seriously going wrong at the white polygons left of centre. The height changes don't match at all. In A the original elevation in this area is around 5.6 m, in B those two white polygons appear to be at 6.4, so this should yield a height change of 0.9 (red) in C. Or are these polygons areas with No Data? If so the colour scales need to be completely different so as to avoid white being part of the scale (so that it can then indicate no-data). Fig7c: show contour lines of height changes beyond significant (as basis for additional clusters?) Stable points not shown? But not clear then how much of area has been allocated properly?

– Indeed the choice of colour scale was not very suitable here. White represents no data, as well as stable areas, as well as the 'stable cluster'. This will be fixed in the future version. The polygons appear as a shadow of the sand pile, where the laser scanner cannot record data.

L304: why six? Not enough justification. What is 'good'? Why not 8 or 10? Isn't there a statistic to tell you when to stop clustering?

– See my second comment in the beginning. We will address this in the new version. (k = 10 gives indeed some more details that are not detected with k = 6.)

Figure 9b: needs horizontal gridlines; labels should be added to lines, rather than a legend (because the sequence in the legend doesn't match the sequence from top to bottom). Vertical axis labels don't make sense: I don't understand how you can have real values asl here for the centroids when the original time-series was mean-subtracted?

– Yes, labels will be added instead of the legend. See previous comments.

L341: this is the only place where you really say that there are no results like 4.2 for the range data; this needs to be made more explicit in Line âĹij300

– obsolete with removal of processing of range time series.

L372-374: please elaborate a bit on this here, please summarize or give us a taste of the cause for this 'curse'.

– more explanation will be added.

5.2: it only gets clear to me here that you use Euclidean for k-means and aggregation, and corr for dbscan! Conclusion: not really clear what all this work benefits; if you remove the clusters associated with the bulldozer work you basically end up with 4 obvious trends: stable, erosion, accretion, noise. This is a bit underwhelming. . .

– see previous comments.

**ESurfD**
[Figure]

**elevation day 1**

North

sandy beach

dunes

test area

paved paths

$(x_t, y_t)$

x [m]

y [m]

elevation [m]

**Fig. 1.** update of Figure 1

---

## Author Response (AR2)

[revised manuscript text omitted]

**Editor's comments**

Thank you for responding to the review comments and suggestions with this comprehensive revision of the paper. I believe it has improved the relevance and impact of this study. In particular, your expansion of presenting results also for k=10 and the additional insights this generates are very useful. I notice, however, that the conclusion has not been edited/revised from the previous version and that the abstract too only contains a few minor text edits. Given that the revised version now presents additional findings and discussion I believe it would be worth your while to edit the abstract and conclusion perhaps a bit more so that they also refer to these new additional findings; otherwise these additional insights may not get picked up by readers as much as you'd like. I am happy to proceed with publication of the current revision if you want, but I herewith want to give you the opportunity to edit the abstract and conclusion if you so wish.

*In response to the editor's comments one sentence has been added each to the abstract and the conclusions.*